# Low-Temperature Fabrication of Plate-like α-Al_2_O_3_ with Less NH_4_F Additive

**DOI:** 10.3390/ma16124415

**Published:** 2023-06-15

**Authors:** Haiyang Chen, Bin Li, Meng Liu, Xue Yang, Jie Liu, Tingwei Qin, Zejian Xue, Yun Xing, Junhong Chen

**Affiliations:** 1School of Materials Science and Engineering, University of Science and Technology Beijing, Beijing 100083, China; chenhaiyang_123@163.com (H.C.); 17801052278@126.com (M.L.); liujie9697@163.com (J.L.); 18511074792@163.com (T.Q.); xue1998829@outlook.com (Z.X.); xingzhyun@163.com (Y.X.); 2School of Civil and Engineering, Hebei University of Architecture, Zhangjiakou 075000, China; yangxue19930817@163.com

**Keywords:** plate-like α-Al_2_O_3_, low temperature, oxalic acid, NH_4_F, synergistic effect

## Abstract

Fluorinated compounds are effective mineralization agents for the fabrication of plate-like α-Al_2_O_3_. However, in the preparation of plate-like α-Al_2_O_3_, it is still an extremely challenging task to reduce the content of fluoride while ensuring a low synthesis temperature. Herein, oxalic acid and NH_4_F are proposed for the first time as additives in the preparation of plate-like α-Al_2_O_3_. The results showed that plate-like α-Al_2_O_3_ can be synthesized at a low temperature of 850 °C with the synergistic effect of oxalic acid and 1 wt.% NH_4_F. Additionally, the synergistic effect of oxalic acid and NH_4_F not only can reduce the conversion temperature of α-Al_2_O_3_ but also can change the phase transition sequence.

## 1. Introduction

Alumina (Al_2_O_3_) is a prospective material due to its competitive thermochemical and mechanical features [1,2,3,4]. Among the dozen crystal structures in which Al_2_O_3_ exists, α-Al_2_O_3_ is the most thermodynamically stable phase, while the others are metastable transition phases [5]. Since α-Al_2_O_3_ is the only thermodynamically stable phase, the preparation of α-Al_2_O_3_ powders with various morphologies, including spherical, plates and fibers, has been sought by many scholars [1,6,7,8,9]. Among these morphologies, the plate-like α-Al_2_O_3_ powder can be applied as a filler for the thermal conductivity improvement of plastics, as a reinforcement for ceramic materials, and as an abrasive for polishing [10,11,12,13,14,15].

Until now, many strategies have been attempted to synthesize plate-like α-Al_2_O_3_, such as hydrothermal methods, melt salt methods, and chemical vapor deposition [13,16,17,18]. Regardless of the methods used, calcination is a necessary step for the fabrication of plate-like α-Al_2_O_3_. Since the phase transition from transitional alumina to α-Al_2_O_3_ involves drastic bond breaking and remaking, the thermal process usually requires temperatures greater than 1200 °C [19]. These temperatures not only impose higher requirements for the preparation equipment of α-Al_2_O_3_, but they also cause substantial mass transfer, leading to rapid growth of α-Al_2_O_3_ particle size [20]. Worse, when the sintering temperature exceeds the critical value, plate-like α-Al_2_O_3_ perhaps converts into particle-like grains [8]. Therefore, it is critical to reduce the synthesis temperature of α-Al_2_O_3_.

It is well known that fluorine-containing compounds can significantly reduce the synthesis temperature of α-Al_2_O_3_, and they are also considered effective mineralization agents for the fabrication of plate-like α-Al_2_O_3_ [5,7,8,14,21,22,23,24,25,26,27,28,29,30,31]. Hence, coupled with the low cost and simple process of α-Al_2_O_3_, the addition of fluorine-containing mineralization agents to alumina precursors has been used by many researchers for the preparation of plate-like α-Al_2_O_3_. For example, Tian et al. [8] synthesized highly pure plate-like α-Al_2_O_3_ from a boehmite precursor with 5 wt.% NH_4_F. Similarly, Wang et al. [21] also successfully prepared plate-like α-Al_2_O_3_ by thermal decomposition of ammonium aluminum carbonate hydroxide (AACH) adding 5 wt.% AlF_3_. They thought that the formation of the intermediate compound AlOF accelerated the mass transfer of Al^3+^ and improved the growth rate along the crystal orientation of <1010>. Nevertheless, a shortcoming of these studies was the excessive use of F content, even as high as 20 wt.% in some cases [31]. When the concentration of F is excessive, both gaseous and particulate fluoride is toxic to plants and animals, and a series of diseases, such as fluorosis of bone and cancer, can be caused [24,32]. In this context, an extremely challenging objective is finding an effective strategy to not only reduce the concentration of F but also to maintain a low synthesis temperature in the process of preparing α-Al_2_O_3_ with fluoride as an additive. Currently, increasing interest has been focused on this aspect, but the existing research has been mostly theoretical, with almost no experimental research.

In our previous report [33], oxalic acid was confirmed to be helpful in reducing the resultant temperature of α-Al_2_O_3_ by 100 °C. Therefore, in this paper, we proposed for the first time adding oxalic acid and NH_4_F as additives in the preparation process of plate-like α-Al_2_O_3_. The synergistic influences of oxalic acid and NH_4_F on the crystal transition order of Al(OH)_3_ were studied, and the optimum F content was explored. Simultaneously, the action mechanism of oxalic acid and NH_4_F during the transformation was clarified based on the experimental results.

## 2. Experimental Procedures

The preparation process mainly involved the following starting materials: analytically pure Al(OH)_3_ (>99.8%), analytically pure oxalic acid (>99.0%), analytically pure ammonium fluoride (NH_4_F, >98.0%), and deionized water. Except for deionized water, which was home made in the laboratory, other raw materials were obtained from Sinopharm Chemical Reagent Co., Ltd., Shanghai, China. All raw materials were directly adopted without any further purification.

As in our previous work [33], first, oxalic acid solution with a concentration of 0.16 mol/L was prepared. Subsequently, 20 g of Al(OH)_3_ and 60 mL of oxalic acid solution were mixed and ball-milled for 11 h. Then, 0, 0.25, 0.5, 1, and 2 wt.% NH_4_F by weight of Al(OH)_3_ was added to continue the ball milling for 1 h. Afterward, the resulting mixtures were air dried at 50 °C for 72 h. The obtained samples were labeled A1 (0 wt.%), A2 (0.25 wt.%), A3(0.5 wt.%), A4 (1 wt.%), and A5(2 wt.%), according to the amount of NH_4_F. Finally, specimens A1–A5 were heated from room temperature to distinct temperatures (700, 750, 800, 850, 900, and 950 °C) at a heating rate of 5 °C/min holding for 3 h.

Phase composition analysis was performed with a D/max2500 X-ray diffractometer (Rigaku Industrial Corporation, Tokyo, Japan) using Cu Kα radiation (40 kV, 40 mA) at 2θ angles of 10–90°. Fourier transform infrared (FTIR) spectroscopy (Nicolet 380) was characterized using a spectrograph (Thermo Electron, Houston, TX, USA). A Nova Nano 450 scanning electron microscope (SEM, FEI Company, Hillsboro, OR, USA) was used to examine the microscopic morphologies of all specimens. X-ray photoelectron spectroscopy (XPS) was performed using an Escalab 50XI photoelectron spectrometer (VG Scientific Ltd., St Leonards, UK). Particle size distribution was collected by a Malvin laser particle size analyzer (Mastersizer 3000, Marvin Pan-alytical, Almelo, The Netherlands). Thermal gravimetry and differential scanning calorimetry (TG-DSC) curves were collected using a simultaneous TG-DTA/DSC apparatus (STA2500, NETZSCH-Gerätebau GmbH, Free State of Bavaria, Germany) under an ambient atmosphere with a heating rate of 10 °C/min up to 1300 °C.

## 3. Results and Discussion

Since the analytic results of A1 were applied in our previous reports [33], only A2–A5 were analyzed in this paper. To characterize the compositions of A2–A5, the spectra of A2–A5 were recorded using the FTIR technique, as shown in Figure 1. Figure 1a exhibits the FTIR spectra in the infrared region of 400–4000 cm^−1^. It was clear that the absorption peaks exhibited by A2–A5 were greatly similar to those of A1 reported in our previous work [33]. The only difference was that, with the increase in NH_4_F content, the absorption peak at around 3200 cm^−1^, corresponding to the stretching vibrations of N-H bonds, began to gradually appear, especially for A5. Among these absorption bands presented by A2–A5, the absorption bands centered at 3621, 3526, 3456, and 3393 cm^−1^ can be assigned to the stretching vibrations of hydroxyl groups in Al(OH)_3_ and adsorbed water [34]. The intensive bands shown at 1704, 1415, and 1297 cm^−1^ were related to the stretching vibration of carboxyl COO^−^, implying that a surface complex (Al_2_C_2_O_8_H_4_), as we reported previously, had been formed due to the interaction of oxalic acid and Al(OH)_3_ [33]. Regarding the bands in the 400 to 1200 cm^−1^ interval (local amplification in Figure 1b), the weak band observed at 915 cm^−1^ was caused by the deformation of surface hydroxyls. The vibration bands at 1022, 970, 803, 751, 666, 560, 517, 451, and 423 cm^−1^ were associated with the stretching and bending of Al-O bonds [35]. With the target of understanding the effects of F^−^ ion on the structure of Al(OH)_3_, the changes in the spectra attributed to Al-O bonds between A2 and A5 were carefully observed. Interestingly, these vibration bands were identical in position and shape except for the intensity, revealing that F^−^ ions did not bond with Al^3+^ ions and O^2−^ ions on the surface of Al(OH)_3_, perhaps because of the surface of Al(OH)_3_ was covered with hydroxyl groups.

The SEM pictures of A2–A5 are illustrated in Figure 2. As shown in Figure 2a, the microstructure of A2 mainly consisted of granular structures, in which a small number of sheets were found, as shown by the red arrow. The formation process of granular particles was discussed in our previous research [33]. As the content of NH_4_F increased, the microstructures of A3-A5 exhibited little change compared with those of A2 (See Figure 2b–d). The illustration in Figure 2d perfectly presents the morphology of granular particles. 

Figure 3a–d demonstrates the XRD patterns of A2–A5 thermally treated at different temperatures holding for 3 h. Learning from the literature, α-Al_2_O_3_ is usually obtained through a sequence of polymorphic transformations because Al_2_O_3_ exists in several different metastable phases. From Figure 3a, it can be clearly observed that, at 700 and 750 °C, the XRD pattern of A2 presented four peaks at 2θ = 37.3°, 42.8°, 46.3°, and 67.0°, corresponding to χ-Al_2_O_3_ and κ-Al_2_O_3_ [5]. For calcination at 800 °C, except for the peaks emerging at 700 °C, some shallow peaks at 2θ = 25.6°, 35.1°, 43.3°, and 57.5°, indexed as α-Al_2_O_3_, can be detected. As the temperature increased to 850 °C, instead of the peaks of χ-Al_2_O_3_, many characteristic peaks ascribed to κ-Al_2_O_3_ can be observed, and κ-Al_2_O_3_ became the major phase coexisting with α-Al_2_O_3_. With the further increase in calcination temperature, the peak intensity of κ-Al_2_O_3_ gradually decreased. At 950 °C, the peak of κ-Al_2_O_3_ disappeared, and the pattern only had the peak of α-Al_2_O_3_. From the above results, for sample A2, the phase transition process was: Al(OH)_3_ → χ-Al_2_O_3_ → κ-Al_2_O_3_ → α-Al_2_O_3_; and the synthetic temperature of α-Al_2_O_3_ was 950 °C, in agreement with our previous study [33]. Regarding A3-A5, the synthesis temperatures of α-Al_2_O_3_ were 900, 850, and 800 °C, respectively, which were all lower than those of A1 and A2. In addition, with the growth of NH_4_F content, χ- and κ-Al_2_O_3_ were replaced by γ- and θ-Al_2_O_3_ in the polycrystalline transformation of Al_2_O_3_, respectively. Surprisingly, except for γ-Al_2_O_3_, no trace of other transitional forms of Al_2_O_3_ were detected before α-Al_2_O_3_ was obtained for A5.

Aiming to further study the influences of oxalic acid and NH_4_F on the transformation sequence of α-Al_2_O_3_, specimen A5 was fired at 800 °C with a heating rare 5 °C/min for different dwell times (10, 30, 60, and 120 min). The XRD images are illustrated in Figure 4. As shown in Figure 4a, the intense diffraction patterns attributed to θ-Al2O3 were observed for A5 thermally treated at 800 °C for 10 min, as well as some shallow patterns indexed as α-Al_2_O_3_. With the prolongation of calcination time, α-Al_2_O_3_ gradually became the main phase. Moreover, the grain size of as-received α-Al_2_O_3_ also displayed an upward trend according to the Scherrer formula, as illustrated in Figure 4b. By combining the results obtained from Figure 3c,d and Figure 4, the phase transformation sequence of A4 and A5 was Al(OH)_3_ → γ-Al_2_O_3_ → θ-Al_2_O_3_ → α-Al_2_O_3_, which was different from that of A1 and A2.

S. J. Wilson [36] suggested that the difference in polycrystalline transition sequences was usually related to the difference in starting materials. In accordance with the results of FTIR, the crystal structure of Al(OH)_3_ was not altered by the introduction of NH_4_F. Therefore, it can be speculated that the structure of the reagent changed during the sintering process. After all, NH_4_F had a low decomposition temperature, beyond which NH_4_F decomposed into NH_3_ and HF gas. The presence of HF gas perhaps plays a significant role in the structural changes of the sintered materials. Figure 5 demonstrates the high-resolution XPS spectra of the transitional Al_2_O_3_ received from A5 at a calcination temperature of 500 °C. Notably, in addition to the characteristic peaks of Al 2p, Al 2s, and O 1s as expected, an exclusive peak ascribed to F 1s can also be detected (Figure 5a). The O 1s region can be attributed to the contributions of the binding energy of O-Al at 530.8 eV and Al-O-H at 532.4 eV (Figure 5b). However, the binding energy of O-Al and Al-O-H exhibited here was higher than that reported in the other literature, which may be caused by the existence of F. With respect to Al 2p spectrum, two peaks centered at 74.2 eV and 75.6 eV, corresponding to the binding energy of Al-O and Al-F, respectively were observed (Figure 5c). Accordingly, the individual line presented one peak at 685.7 eV belonging to the binding energy of F-Al (Figure 5d). The results of XPS confirmed that F existed on the surface of the calcined materials. Adopting Al(OH)_3_ as a starting material, the first metastable phase received was χ-Al_2_O_3_, which inherited the arrangement of oxygen anions in the crystal structure of Al(OH)_3_, and Al^3+^ ions occupied the octahedral position in the hexagonal oxygen layer. In comparison, the stacking of χ-Al_2_O_3_ in the c direction demonstrated extremely obvious disorder [37,38]. Consequently, it turned out that Al(OH)_3_ showed a significant disorder along the c axis during its dehydration, which may be an opportunity for gaseous HF to change the structure of the starting regent in calcination. The gaseous HF working with Al^3+^ ions on the surface of the precursor in calcination might cause the collapse of the disordered Al-O octahedron, resulting in the formation of γ-Al_2_O_3_ so that the polycrystalline transformation sequence changed to Al(OH)_3_ → γ-Al_2_O_3_ → θ-Al_2_O_3_ → α-Al_2_O_3._

Figure 6 shows the TG-DSC curves of Al(OH)_3_, A1, and A5. As presented in Figure 6a, the TG-DSC curves of Al(OH)_3_ can be divided into two stages. The first stage, with a mass loss of 33.8% exhibited, two endothermic peaks at about 85 and 292 °C. These endothermic peaks were related to the evaporation of physically adsorbed water and the decomposition of Al(OH)_3_. In the second stage, weight loss of around 2% can be observed, corresponding to the removal of hydroxyl groups from the transition Al_2_O_3_ surface. Furthermore, two exothermic peaks were displayed at 870 and 1120 °C, which were caused by the appearance of κ- and α-phases, respectively [39]. Similar to Al(OH)_3_, the TG-DSC curves of A1 also included two steps, two endothermic peaks and two exothermic peaks, as shown in Figure 6b. The difference was that the two endothermic peaks were decreased to 800 and 1050 °C, respectively, and the heat release increased when the α-phase appeared. The growth in heat release indicated that the nucleation amount of α-Al_2_O_3_ increased. There were two main reasons for the decrease in the phase transition temperature and the increase in the nucleation number. One was that the complex generated by Al(OH)_3_ and oxalic acid can act as seeds, providing low energy positions for heterogeneous nucleation. On the other hand, at greater than 400 °C, the number of transition Al_2_O_3_ surface hydroxyl groups obtained from A1 was approximately twice that of the transition Al_2_O_3_ obtained from Al(OH)_3_, which can be obtained from the mass loss in the second stage of TG-DSC. Of course, this outcome also indicated that, at greater than 400 °C, A1 released twice as many water molecules as Al(OH)_3_ released. As is known, the size of metastable Al_2_O_3_ must reach a critical crystal diameter before converting to α-Al_2_O_3_ [40]. The water vapor produced by dehydroxylation contributes to the coarsening of metastable Al2O3 to reach the critical crystal size [41]. Moreover, the formation of vacancies by dehydroxylation is also beneficial to the rearrangement of the Al_2_O_3_ lattice structure [42]. Therefore, A1 is more easily converted into α-Al_2_O_3_ than Al(OH)_3_. In contrast with Al(OH)_3_ and A1, for the TG-DSC curves of A5, the exothermic peak attributed to α-Al_2_O_3_ nucleation that we cared about decreased to 940 °C and was more intense, related to the mass transfer of AlOF gas, as well as the reasons mentioned for A1. Additionally, C. Barth et al. [43] believed that, once α-Al_2_O_3_ at a high temperature was exposed to water, Al(OH)_3_ microcrystalline clusters would form on the surface of α-Al_2_O_3_. In our work, the XPS of transition Al2O3 mentioned above confirmed the presence of Al-F bonds, which might produce AlF_3_ clusters at the position of Al-F bonds, providing nucleation sites for α-Al_2_O_3_, which was also responsible for lowering the resultant temperature of α-Al_2_O_3._ In short, α-Al_2_O_3_ can be obtained at a low temperature under the synergistic effects of oxalic acid and NH_4_F. Nevertheless, the synthesis temperature of α-Al_2_O_3_ obtained by the results of TG-DSC was higher than that shown by XRD, which may be caused by the high heating rate and purging of air during the operation process of TG-DSC.

Figure 7 shows SEM images of α-Al_2_O_3_ obtained from A2–A5 at different sintering temperatures. As can be seen from Figure 7a, the produced α-Al_2_O_3_ of A2 at 950 °C displayed a granular microstructure. Different from A2, at the same temperature, a drum-like architecture of α-Al_2_O_3_ particles was obtained for A3, while the α-Al_2_O_3_ prepared by A4 and A5 presented a plate-like structure, as illustrated in Figure 6b,c. Meanwhile, by carefully comparing the obtained α-Al_2_O_3_ in Figure 6b,c, it can be seen that the particle diameter of α-Al_2_O_3_ gradually increased with the increase in NH_4_F content, while the thickness exhibited a decreasing trend. From the perspective of crystal growth, the microstructure of materials depends on the growth rate of the crystal plane. The remaining crystal surface is a plane with a slow growth rate. The inset in Figure 6b perfectly exhibits the crystal plane of α-Al_2_O_3_ obtained from A3 at 950 °C. According to a study [44], the side of the drum-shaped α-Al_2_O_3_ corresponds to the crystal plane of (2243), and the front side belongs to the (0001) plane. Different from the side of drum-shaped α-Al_2_O_3_, the side of the plate-like structure synthesized from A4 and A5 was ascribed with an (1120) plane. It was reported that F^−^ ions can work with Al^3+^ ions and adsorb on the crystal face of α-Al_2_O_3_. In α-Al_2_O_3_ unit cells, the (0001) plane has close-packed atoms, lending themselves to more bare Al^3+^ ions on this crystal face. As a result, F^−^ ions tended to adsorb on the (0001) plane, limiting the deposition of α-Al_2_O_3_ on this face and bringing down its growth rate, resulting in the retention of the (0001) plane [21]. Of course, this was also responsible for the larger size and thinner thickness of α-Al_2_O_3_ obtained in the case of higher NH_4_F content. In general, a small primary grain size was helpful in improving the mechanical properties of ceramic bodies. Figure 7e,f illustrates the SEM images of α-Al_2_O_3_ synthesized at 850 °C from A4 and at 800 °C from A5. Apparently, the produced α-Al_2_O_3_ presented similar particle sizes. Considering the particle size of as-received α-Al_2_O_3_, the F content, and the resultant temperature, under the synergistic effects of oxalic acid and NH_4_F, mixing 1 wt.% NH_4_F was optimal. In summary, plate-like α-Al_2_O_3_ can be successfully synthesized by the incorporation of 1 wt.% NH_4_F with the assistance of oxalic acid at 850 °C. Table 1 illustrates detailed information about the synthesis of α-Al_2_O_3_ when fluorine-containing mineralizing agents are used. Compared with other reports, the fluoride content and synthesis temperature in this study were the lowest when the morphology of as-synthesized α-Al_2_O_3_ was plate-like, confirming the validity of our work.

## 4. Conclusions

In the current work, oxalic acid and NH_4_F were proposed for the first time as additives in the preparation of plate-like α-Al_2_O_3_. The results showed that plate-like α-Al_2_O_3_ can be synthesized at a low temperature of 850 °C with the synergistic effects of oxalic acid and 1 wt.% NH_4_F. Both the synthesis temperature and the content of fluoride were lower than in other reports. With the increase in NH_4_F content, due to the adsorption of F^−^ and the formation of AlOF gas, the morphology of the as-obtained α-Al_2_O_3_ changed from granular to plate-like, and the resultant temperature presented a downward trend. Additionally, owning to the adsorption of HF on the starting materials in sintering, the transition order of α-Al_2_O_3_ was changed from Al(OH)_3_ → χ-Al_2_O_3_ → κ-Al_2_O_3_ → α-Al_2_O_3_ to Al(OH)_3_ → γ-Al_2_O_3_ → θ-Al_2_O_3_ → α-Al_2_O_3_.

## Figures and Tables

**Figure 1 materials-16-04415-f001:**
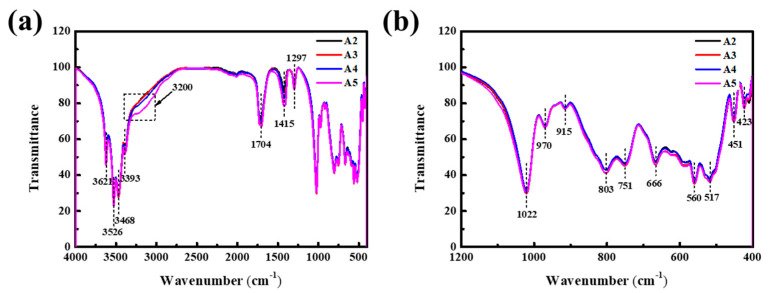
FTIR spectra of: (**a**) the specimens A2–A5; (**b**) local amplification at 1200–400 cm^−1^.

**Figure 2 materials-16-04415-f002:**
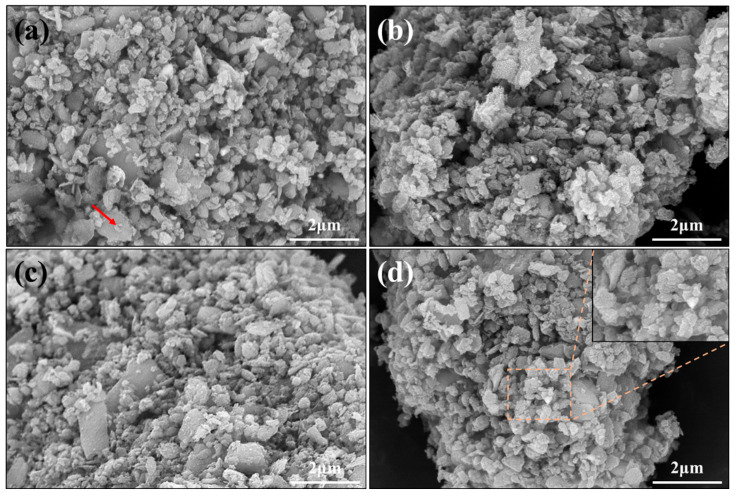
SEM pictures of A2–A5: (**a**) A2; (**b**) A3; (**c**) A4; (**d**) A5.

**Figure 3 materials-16-04415-f003:**
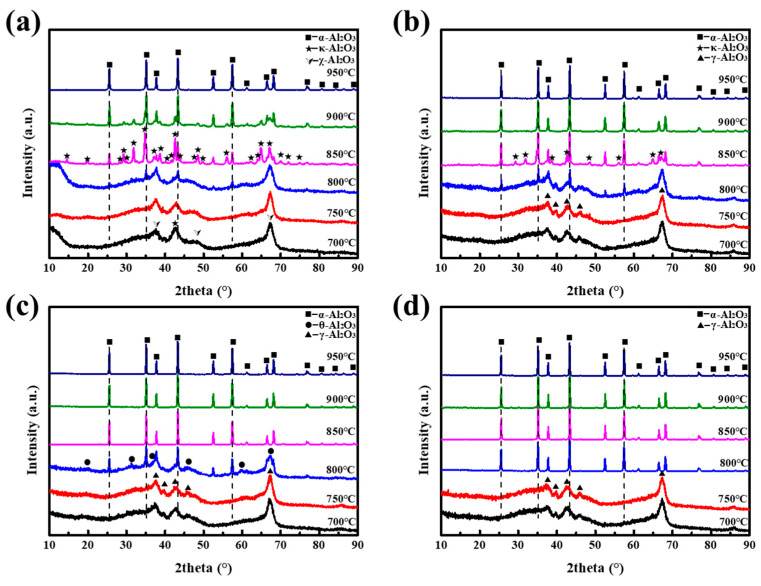
XRD patterns of the alumina phases received from A2–A5 at different calcination temperatures: (**a**) A2; (**b**) A3; (**c**) A4; (**d**) A5.

**Figure 4 materials-16-04415-f004:**
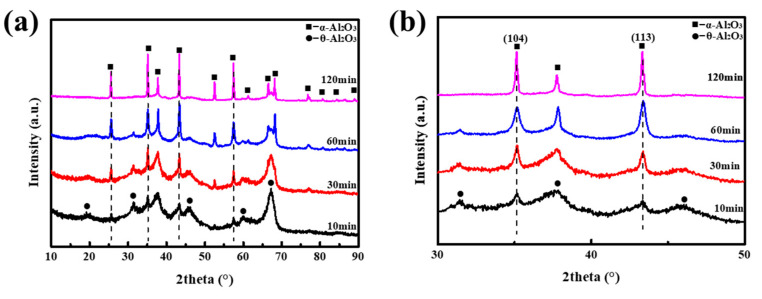
XRD patterns of: (**a**) the transition Al_2_O_3_ as obtained from A5 at 800 °C for distinct times; (**b**) local amplification at 2θ = 30°–50°.

**Figure 5 materials-16-04415-f005:**
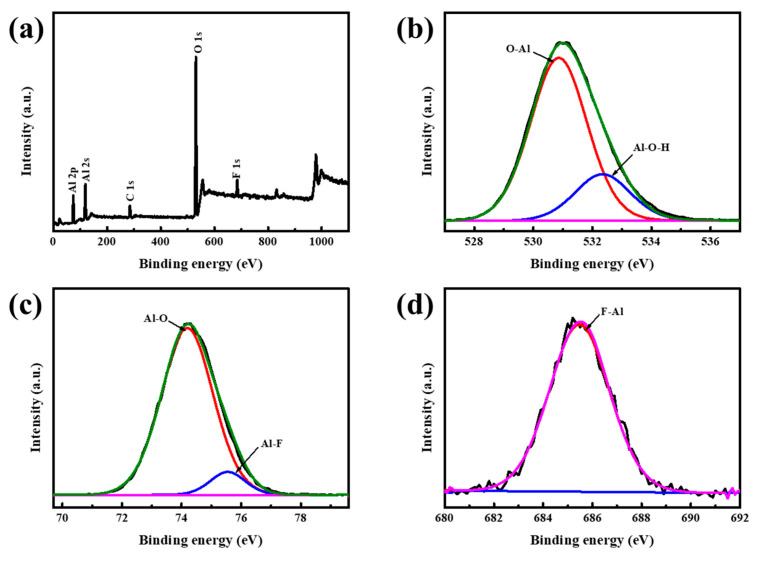
XPS spectrum of transition Al_2_O_3_ as-obtained from A5 at 500 °C (**a**); the overall XPS spectrum and the individual lines of: (**b**) O 1s; and (**c**) Al 2p (**d**) F 1s.

**Figure 6 materials-16-04415-f006:**
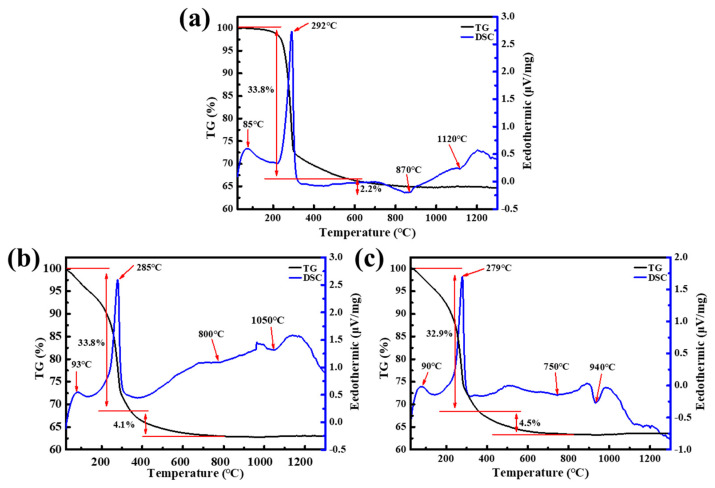
TG-DSC curves of: (**a**) Al(OH)_3_; (**b**) A1; (**c**) A5.

**Figure 7 materials-16-04415-f007:**
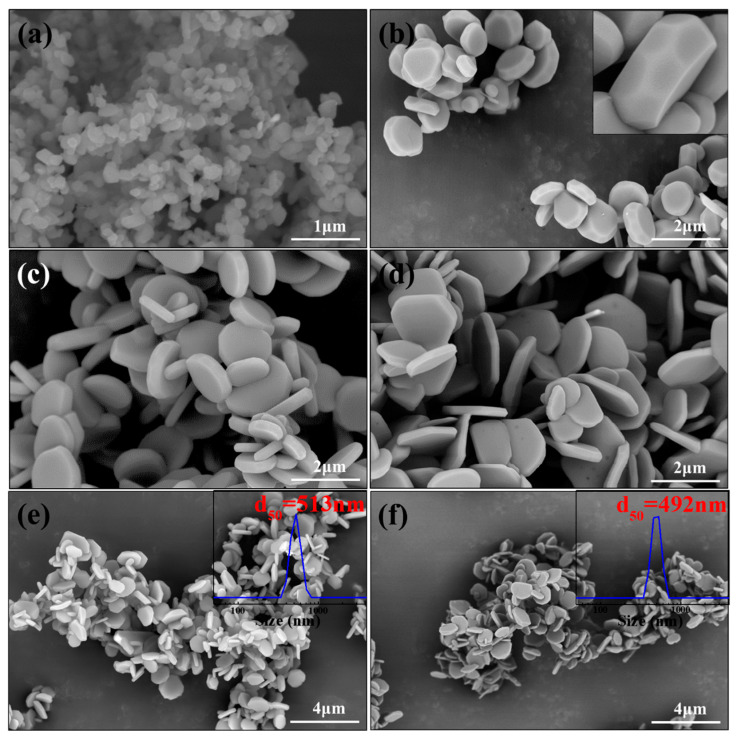
SEM images of α-Al_2_O_3_ obtained from A2–A5 at different temperatures: (**a**) A2-950; (**b**) A3-950; (**c**) A4-950; (**d**) A5-950; (**e**) A4-850; (**f**) A5-800.

**Table 1 materials-16-04415-t001:** Detailed information about the synthesis of α-Al_2_O_3_ with fluoride as additives.

Morphology	Additive	Mixing Amount	Material	Temperature/Time	Ref.
Polyhedral	AlF_3_	1 wt.%	Al(OH)_3_	910 °C/3 h	[5]
Sphere-like	NH_4_BF_4_	5 wt.%	Al(OH)_3_	1450 °C/3 h	[7]
Plate-like	NH_4_F	5 wt.%	AlOOH	900 °C/2.5 h	[8]
Plate-like	NH_4_F+NH_4_F	5 wt.% + 5 wt.%	Al(OH)_3_	1300 °C/3 h	[14]
Plate-like	AlF_3_	5 wt.%	AACH	1200 °C/1 h	[21]
Plate-like	AlF_3_	2 wt.%	AIP	900 °C/3 h	[22]
Plate-like	AlF_3_ NH_4_F	2.8 wt.% of F	AlOOH	1050 °C/1 h 950 °C/1 h	[23]
Plate-like	AlF_3_	0.6 wt.%	t-Al_2_O_3_Al(OH)_3_	1000 °C/0.5 h 950 °C/0.5 h	[24]
Plate-like	AlF_3_	1 mol%	Al(OH)_3_	750 °C/10 h	[25]
Plate-like	AlF_3_	2 mol%	AlOOH	750 °C/10 h	[26]
Plate-like	ZnF_2_ AlF_3_	-	Al(NO_3_)_3_	920 °C/10 h 900 °C/1 h	[28]
Square-like	NH_4_F	2 wt.%	γ-Al_2_O_3_	1300 °C/2 h	[30]
Plate-like	Oxalic acid + NH_4_F	0.16 mol/L + 1 wt.%	Al(OH)_3_	850 °C/3 h	This work

## Data Availability

No new data were created or analyzed in this study. Data sharing is not applicable to this article.

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
