# Peer review of "Low-Temperature Fabrication of Plate-like α-Al2O3 with Less NH4F Additive"

_materials, 2023, doi:10.3390/ma16124415_

Round 1
Reviewer 1 Report
In this manuscript, the results of this research are conveyed thoughtfully and completely, and they are consistent with the experimental findings. However, the authors failed to explain and draw out the novelty of the work, this aspect needs to be improved. This work is worthwhile to be publish in this journal after minor revision. The following issues should be addressed:
1. Introduction is well-organized but the importance and novelty of the research should be highlighted and more clearly stated. The authors should give some examples of works in the bibliography, to clear the advantage of their work in comparison with those works.
2. The authors are responsible for the English, which should be polished throughout the manuscript to clear some minor typo/grammar errors.
3. Introduction part, if possible, some important and relative reports references could help:
10.1016/j.ceramint.2021.03.223
10.1134/S0036023620040099
Hence, I recommend it accepted for publication after minor revisions.
Reviewer 2 Report
1. The process of synthesis of α-Al2O3 has been well studied for a long time. The introduction should reflect the work on the thermal treatment of nanoboehmite (10.1134/S0036023620040099) and the hydrothermal synthesis of boehmite of various morphologies (10.1134/S0036023621030104; 10.1134/S0036024415040196; 10.1134 /S0040579513040143)
2. In the introduction, it is necessary to describe in detail the results on the use of mechanical activation in the preparation of α-Al2O3
3.If you indicate the presence of adsorbed water, then there should be a band at 1645 Sm-1 in the IR spectrum. However, she is missing. What does it have to do with?
4. Fig. 1 and Fig. 3 are small and unreadable.
5.What is the reason for the absence of the γ-Al2O3 phase for sample A2 in the scheme: Al(OH)3 → χ-Al2O3 → κ-Al2O3 → α-Al2O3
6. Fig. 3- Which of the diffraction patterns corresponds to samples A2, A3, A4, A1
7. Fig. 6. What is the surface water content?
Minor editing of English language required
Reviewer 3 Report
Manuscript materials-2389880
(Comments to Both Author and Editor):
Authors tried to show a new method to fabricate disk-shaped a-Al2O3 at relatively low temperature.
First of all, authors should have prepared the manuscript more carefully. For example, many subscriptions and superscriptions were not used, and English should have been corrected such as Line 36 and Line 73 before submitting the manuscript.
Experimental section should have been prepared more carefully as well.
Line 69: Any purity information?
Line 73: How was the weight % of NH4F was calculated if that was added to a solution with a certain volume?
Line 77: Drying condition: Aerobic, anaerobic, or etc.?
Results section should have been also improved.
Lines 92 115: Why not A1, which should serve as a control?
Line 116: The sheets need to be pointed in Fig. 2.
Line 122: Too little caption.
Line 138: A1 is not shown in Figs. 2-4.
Line 143: Which is which sample?
Line 229: Any particle size analysis?
Line 255: Based on the photos, it would be better to describe the shape as disk rather than plate.
Line 256: (5) -> (d)
Low. The Authors should have improved before submission.
